# Using specialist screening practitioners (SSPs) to increase uptake of bowel scope (flexible sigmoidoscopy) screening: results of a feasibility single-stage phase II randomised trial

Lesley M McGregor,[1] Hanna Skrobanski,[1,2] Mary Ritchie,[3] Lindy Berkman,[4] Hayley Miller,[3,5] Madeleine Freeman,[1] Nishma Patel,[6] Stephen Morris,[6] Colin Rees,[7] Christian von Wagner[1]

For numbered affiliations see end of article.

**Correspondence to**
Dr Lesley M McGregor;
l.mcgregor@ucl.ac.uk

## ABSTRACT

**Objective** To determine the feasibility of specialist screening practitioners (SSPs) offering patient navigation (PN) to facilitate uptake of bowel scope screening (BSS) among patients who do not confirm or attend their appointment.

**Design** A single-stage phase II trial.

**Setting** South Tyneside District Hospital, Tyne and Wear, England, UK.

**Participants** Individuals invited for BSS at South Tyneside District Hospital during the 6-month recruitment period were invited to participate in the study.

**Intervention** Consenting individuals were randomly assigned to either the PN intervention or usual care group in a 4:1 ratio. The intervention involved BSS non-attenders receiving a phone call from an SSP to elicit their reasons for non-attendance and offer educational, practical and emotional support as required. If requested by the patient, another BSS appointment was then scheduled.

**Primary outcome measure** The number of non-attenders in the intervention group who were navigated and then rebooked and attended their new BSS appointment.

**Secondary outcome measures** Barriers to BSS attendance, patient-reported outcomes including informed choice and satisfaction with BSS and the PN intervention, reasons for study non-participation, SSPs' evaluation of the PN process and a cost analysis.

**Results** Of those invited to take part (n=1050), 152 (14.5%) were randomised into the study: PN intervention=109; usual care=43. Most participants attended their BSS appointment (PN: 79.8%; control: 79.1%) leaving 22 eligible for PN: only two were successfully contacted. SSPs were confident in delivering PN, but were concerned that low BSS awareness and information overload may have deterred patients from taking part in the study. Difficulty contacting patients was reported as a burden to their workload.

**Conclusions** PN, as implemented, was not a feasible intervention to increase BSS uptake in South Tyneside. Interventions to increase BSS awareness may be better suited to this population.

**Trial registration number** ISRCTN13314752; Results.

### Strengths and limitations of this study

► This study was the first to consider patient navigation as an intervention to increase uptake of bowel scope screening (BSS) as part of the UK National Health Service Bowel Cancer Screening Programme.

► We aimed to use the already advanced knowledge and communication skills of specialist screening practitioners for the delivery of the intervention.

► Contact information obtained by the Bowel Cancer Screening Programme for people to be invited for BSS does not include telephone numbers and, therefore, an opt-in consent process was a necessary design feature for this study.

## INTRODUCTION

Colorectal cancer (CRC) is the second most common cause of cancer death in the UK.[1] In 2013, the National Health Service (NHS) Bowel Cancer Screening Programme (BCSP) introduced bowel scope screening (BSS) for men and women aged 55 years. BSS involves a once-only flexible sigmoidoscopy, a test that can substantially reduce CRC incidence and mortality by finding and removing precancerous polyps. However, the benefits of BSS are dependent on uptake, which is currently less than optimal at 43.1%, a figure that falls further when the focus is on the more deprived areas of England.[2] There is, therefore, a need for interventions to increase BSS uptake and reduce the observed social gradient.

Patient navigation (PN) is an intervention that aims to remove logistical or psychological barriers to healthcare. It involves the provision of one-to-one tailored support from trained individuals (ie, patient navigators) to patients who need it. This may be delivered

face to face or via telephone. The concept of PN was originally developed in the USA, in 1990, by Harold Freeman, mainly in the context of breast cancer screening for patients of low socioeconomic status.[3] However, it has since been applied to other cancer types and chronic diseases across the healthcare continuum, from prevention to end of life.[4]

There is a lot of US evidence that PN can increase CRC screening uptake. A randomised controlled trial showed significantly higher rates of screening (completion of either a faecal occult blood test (FOBT) or colonoscopy) among patients in a PN intervention group (35%), compared with those in a control group (20%).[5] More recently, DeGroff et al[6] found that low-income patients who received a PN intervention were one and a half times more likely to attend colonoscopy than control participants receiving usual care.[6] Similarly, Rice et al[7] found patients assigned to a PN intervention condition were 11.2 times more likely to attend colonoscopy than those in the control condition.[7] Patients in the navigation intervention group were also significantly more likely to have adequate bowel preparation and significantly less likely to miss or cancel their appointments.[7] Although PN has not yet been applied to BSS in the UK, it has successfully been used to increase uptake of breast screening among African-Caribbean women in two socially diverse areas of London.[8] Moreover, a further single-site study in London explored whether a telephone call from a general practitioner or healthcare assistant to patients who had failed to respond to their NHS BCSP invitation to complete an FOBT could help increase screening uptake.[9] The call aimed to identify reasons for their non-response and encourage participation. The results indicated that the call was successful, with 66% of those contacted agreeing to be screened and 50% then completing the test.[9] This suggests that a PN telephone intervention could be an effective method to promote BSS attendance in the UK.

The main aim of this study was to determine the feasibility of employing specialist screening practitioners (SSPs), who are based within screening centres and whose day-to-day role is to support patients through the screening process, to additionally offer PN to patients who do not confirm or attend their BSS appointment. The advantage of delivering PN through screening centres as opposed to primary care was the already available infrastructure to support patients through their screening journey, for example, readily available expertise and knowledge regarding BSS and the BCSP, skills communicating with the public about cancer and screening tests, and the ability to directly coordinate amendments to screening invitations, for example, rescheduling appointments, arranging for the enema to be administered at the hospital. The main potential barrier to delivering PN in this way was that screening centres do not have access to the telephone numbers of screening invitees. The objectives of this study were, therefore, to determine the feasibility of screening centre delivered PN by assessing study recruitment rates (specifically the proportion of screening invitees willing to consent to taking part and sharing their contact details); BSS uptake rates of those navigated; patients' satisfaction with PN; SSPs' satisfaction with PN and the cost of delivering the PN intervention.

## METHOD
### Design
We used a single-stage phase II trial to determine whether PN could positively impact the BSS attendance rate in South Tyneside to a degree that would merit further investigation. The protocol for this research has previously been published and should be consulted for additional methodological detail.[10]

### Recruitment and setting
All individuals invited for BSS at South Tyneside District Hospital (STDH; one of three sites operating under the South of Tyne Screening Centre) during the recruitment period (May to October 2015) were invited to take part in this study. The South of Tyne Screening Centre was chosen because it has below average uptake of BSS, with 37% of invited adults attending screening.[2]

### Procedure
A study invitation was sent with the standard BSS preinvitation letter and included a participant information sheet (PIS) and consent form to be completed and returned by those who wished to participate in the study. The PIS detailed information about the aim of the study, consent process, data protection and study procedure, including the telephone number that would appear on their screen if they were to receive a PN phone call. It also stated that deciding to take part in the study was not the same as deciding to take part in screening. The consent form asked individuals to indicate their gender, contact number, name and address. Individuals who did not want to participate were asked to complete and return an anonymised non-participation postcard (A5 size), using the presented tick box options and free-text space to indicate their reason(s) for non-participation (online supplementary file A).

The research nurse allocated (unblinded) consenting individuals to either the PN intervention or control group using a pregenerated, gender-specific randomisation list (provided by medical statistician). An initial randomisation ratio of 2:1 in favour of the intervention was soon increased to 4:1 due to a low response rate.

As per usual care, 2 weeks after receiving the preinvitation letter, consenting participants received a BSS appointment date and time (6 weeks in advance). This letter requested confirmation of their intention to attend the appointment (return of slip or call to centre). If no confirmation was received within 2 weeks, the bowel cancer screening hub sent a reminder letter. If confirmation was not received within a further 2 weeks, a cancellation letter was then sent. For those in the control group, there was no further contact from the screening centre. If

the appointment was confirmed, an enema with instructions was mailed out approximately 2 weeks before the appointment. Patients who confirmed but then failed to attend received a 'DNA' letter. For those in the control group, again no further contact was made.

For those in the PN group, the PN intervention began following the mail out of the cancellation or 'DNA' letter. In addition to the original protocol, the PN intervention was also initiated following a call made by a patient to the screening centre to cancel their BSS appointment without an intention to rebook at a later date. For those who phoned to cancel an appointment with a plan to call back and rebook, the PN intervention was only initiated if no new appointment was made within 6–8 weeks. As per study protocol, the PN intervention involved an SSP trained in how to deliver PN, telephoning the individual to elicit reasons for non-attendance, to provide emotional and instrumental support to overcome any barriers identified (eg, assistance with the enema), and, if relevant, to help arrange a new appointment.[10] Good practice required the participant to confirm their agreement to the call being recorded at the start of the PN call.

All participants were sent one of four versions of an End of Study Questionnaire (ESQ) approximately 4 weeks after attendance at their BSS appointment, or 1 week after non-attendance or an unsuccessful PN call (ie, no response or no new appointment agreed). Questions varied across versions depending on relevance to study group and attendance status but generally included questions pertaining to demographic details, BSS knowledge and decision experience and satisfaction. For those who attended, additional questions included the test experience and views on the support they received. Specifically, patient-reported pain and embarrassment were explored as these are common barriers to screening uptake.[11–16] The ESQ for patients who received PN also aimed to explore the cognitive and emotional response to the intervention and its effect on informed choice and BSS knowledge. If ESQs were not returned within 2 weeks, a second ESQ was sent as a reminder.

### SSP interviews

The six SSPs (all female) trained to deliver PN were interviewed at the end of the study to elicit their perspectives on the delivery of the intervention, including barriers and facilitators, and how PN would likely affect their workload if implemented. Semistructured telephone interviews were carried out by the patient representative (LB) involved in the study and recorded. Following transcription, the interviews were then analysed for recurrent themes using a thematic analysis.

### Patient involvement

The research question was developed as part of a stakeholder meeting which included three patient representatives. The meeting introduced the PN concept, reviewed previous evidence supporting its use and determined the suitability of this strategy in the English context. The meeting also determined the appropriateness of using a feasibility study, the specific way in which PN should be delivered (eg, by SSPs) and the potential for patient and public involvement (PPI) going forward. One of our attending patient representatives became a coinvestigator and was involved in reviewing drafts of the protocol, recruitment materials, the lay summary, conference abstracts and reports. In addition, our patient representative (LB) went on to help develop the SSP interview schedule, then conducted the interviews, coanalysed the transcripts and disseminated the results at a conference.

### Cost analysis

We undertook a detailed bottom-up microcosting exercise to calculate the average cost per participant of PN and usual care. Costs mainly comprised staff time and stationary (stamps, envelopes, paper, postage and printing charges) associated with the following activities: (1) the preinvitation letter introducing BSS; (2) the invitation letter to participate in BSS with an appointment date; (3) the reminder letter sent if no confirmation is received from the patient; (4) the cancellation letter sent to participants if there is no reply to the reminder letter and (5) instructions on how to participate in BSS sent to participants if confirmation is received. In addition, for PN, we also included the cost of PN training and the cost of PN contacts in the event that patients did not respond to the invitation or did not attend a confirmed appointment. Unit costs were taken from market prices.

## RESULTS

### Response rate

The response rate was 14.6% (153 out of 1050), which was significantly below the target of 40%.[10] One consent form was received after the individual had attended their scheduled BSS appointment and was therefore excluded from randomisation, resulting in 43 participants randomly allocated to the control group (female=23; 53.5%) and 109 to the intervention (PN) group (female=58; 53.2%). The non-randomised individual (female) was included in the evaluation and sent an ESQ.

### Study decliners

Non-participation postcards were returned by 16 people (1.8% of 897 non-consenting invitees). The most commonly reported reasons for non-participation were 'I have already decided not to have the BSS test' (43.8%; n=7), and 'I do not want to receive additional phone calls from the screening centre' (37.5%; n=6). Other reasons for not having the test mostly focused on people currently feeling well/content with life and not wanting to alter this by getting involved in screening.

### BSS participation by study group

The majority of participants attended their original BSS appointment: control=79.1% and PN=79.8%. This level of attendance was much higher than the 35% average for

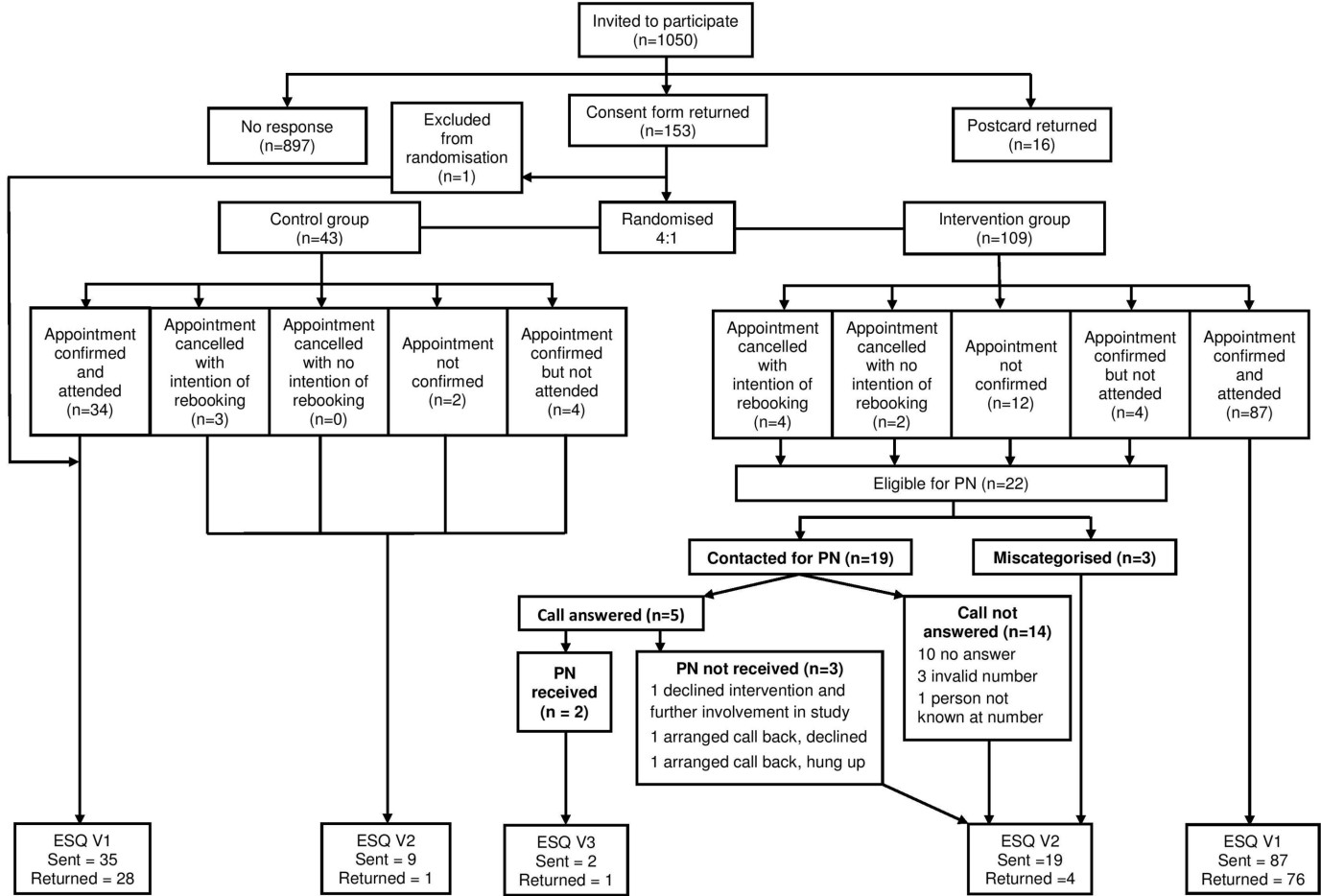

**Figure 1** Flow diagram of study participation. ESQ, End of Study Questionnaire; PN, patient navigation.

STDH over the study period. (Ritchie, personal communication, 2016) As a result, only 22 people were eligible for the intervention: 12 failed to confirm their appointment, 4 failed to attend a confirmed appointment and 6 cancelled their appointment. However, SSPs only called 19 participants as three were lost to follow-up (ie, erroneously not highlighted as needing PN intervention) and only two received PN. The PN received was of a relatively 'low level': one PN call was audio recorded (2 min 54 s), and it revealed that the only reason for non-attendance was that the individual's GP had recently referred them for bowel investigations. For the second PN call, the individual stated that they had forgotten their appointment and asked to rebook before permission to record was obtained. Of the remaining participants, 10 failed to answer all call attempts, 4 had an invalid or wrong number and 3 answered but declined participation (see figure 1).

### Attendance follow-up (intention to treat)
Attendance data were assessed for each consenting individual, 3 months after the last PN call attempt was made. In this time, one control participant had self-referred and attended a new BSS appointment and three PN participants (one failed to answer PN calls, one received a PN call and one was lost within the study and so the PN intervention was not attempted) rebooked and attended a BSS appointment. BSS attendance levels then increased to 81.4% (n=35) and 82.6% (n=90) for the control and PN groups, respectively.

### End of Study Questionnaires
One hundred and ten participants completed ESQs; the majority of these had attended their original BSS appointment (n=104; male=54) and therefore none had received the PN intervention. While comparisons between those who did and did not receive the intervention were not possible, useful data emerged regarding the BSS experience: 98.1% were satisfied with their choice to attend, 97.2% were satisfied with their test experience and 100% were satisfied with the support they received throughout the BSS experience. The majority of participants reported no or mild pain (59.6%) and indicated no or mild embarrassment (78.9%). Furthermore, 30.4% found it less painful and 46.2% less embarrassing than expected. Table 1 shows the breakdown of these responses.

Although the majority of attendees were able to correctly answer 10 knowledge questions presented regarding BSS, there remained a notable percentage of BSS attendees who answered these questions incorrectly or with uncertainty. For example, 23% did not believe that BSS can help prevent bowel cancer and 18% did not consider bowel cancer to be a common cancer. Of

**Table 1** End of Study Questionnaires

| Question | Attenders (n=104)<br>% (n) |
|---|---|
| **I am satisfied with my choice to attend bowel scope screening (BSS)** | |
| Strongly agree | 67.3 (70) |
| Agree | 30.8 (32) |
| Disagree | 1.0 (1) |
| Strongly disagree | 1.0 (1) |
| **Overall, how satisfied were you with your test experience?** | |
| Very satisfied | 63.5 (66) |
| Satisfied | 33.7 (35) |
| Dissatisfied | 1.0 (1) |
| Very dissatisfied | 1.9 (2) |
| **Overall, how satisfied were you with the level of support you received throughout the BSS experience?** | |
| Very satisfied | 88.5 (92) |
| Satisfied | 11.5 (12) |
| Dissatisfied | 0.0 (0) |
| Very dissatisfied | 0.0 (0) |
| **How much pain did you feel during your test?** | |
| None | 17.3 (18) |
| Mild | 42.3 (44) |
| Moderate | 27.9 (29) |
| Severe | 12.5 (13) |
| **Was your test more or less painful than you expected?** | |
| More painful | 23.5 (24) |
| As expected | 45.1 (46) |
| Less painful | 30.4 (31) |
| **How much embarrassment did you feel during your test** | |
| None | 43.3 (45) |
| Mild | 35.6 (37) |
| Moderate | 17.3 (18) |
| Severe | 3.8 (4) |
| **Was your test more or less embarrassing than you expected?** | |
| More embarrassing | 5.8 (6) |
| As expected | 48.1 (50) |
| Less embarrassing | 46.2 (48) |

particular interest was that 60% did not know that BSS was a one-off test.

### SSP interviews

All six SSPs were interviewed (range:19–41 min). Overall, SSPs found the PN training to be very useful and were confident in delivering the intervention. However, they also expressed some concerns about PN and highlighted possible reasons why it was unsuccessful for this patient group. SSPs felt that within South Tyneside, individuals not wanting to attend BSS would also not want to discuss their reasons:

> If they're not going to take part in something then you're probably not going to consent to be contacted after because you've already made your mind up that you're not going to attend. (SSP5)

Additionally, they mentioned that people in their area were largely unaware of BSS, and some would have been anxious on receiving the invitation letters as a consequence:

> I think because people don't really know what bowel scope screening is. I think some people were frightened because they thought that their GPs had referred them in. I think there's not enough information out about bowel scope screening as yet. (SSP4)

SSPs also worried that patients would have been overwhelmed by all the information they received about both BSS and the study simultaneously, and this may have deterred some individuals from participating. This may also have caused confusion about whether patients were consenting to take part in screening or a research study, and perhaps some attended screening under the assumption that it was part of the study:

> I think it was confusing for them. I think they received a lot of information because they were receiving the navigation information and they were also receiving information regarding bowel screening. I think it was probably too much all at one time for them to comprehend. I think it probably put some patients off, and certainly from the experience that I had from consenting people for bowel scope screening. One lady that springs to mind was a school teacher; she thought that she was consenting for a study rather than for a screening investigation. I think it created confusion. (SSP4)

In addition, SSPs expressed difficulty with trying to contact non-responders eligible for PN, and commented that many participants had provided false telephone numbers or did not answer. Those they were able to contact often hung up or seemed uninterested in talking to them. These difficulties in contacting patients meant SSPs needed to repeat phone calls on different days and times, which was perceived as a demanding addition to their workload:

> 'I've been ringing the patients on different days, different times of the day, that sort of thing, to try and… if people are at work and things, to try and catch them when they're there. So, doing it on top [of] my everyday other clinical commitments, it has actually been quite a demand.' (SSP3)

### Cost analysis

The mean costs per participant of PN and usual care were £18.92 and £12.10, respectively. The difference in costs

(£6.72 per participant) was driven mainly by the cost of SSP time associated with trying to contact non-responders.

## DISCUSSION

This study explored the feasibility of SSPs offering PN to patients at South Tyneside Screening Centre who did not confirm or attend their appointment, in order to engage/re-engage them with the opportunity to have BSS. The introduction of PN, with this patient group, was not found to be feasible within the current programme structure.

This was the first study to apply PN to BSS in an area of low uptake in the UK. A range of quantitative and qualitative data was gathered to assess the feasibility of the intervention, including BSS attendance, reasons for study non-participation, informed choice about BSS, an SSP evaluation and a cost evaluation. The study provides important practical implications, because it suggests that SSP-led PN is currently not a feasible method for increasing screening uptake, in areas where BSS is low, predominantly due to restricted access to contact details. A cost analysis also suggested that compared with other interventions aimed at increasing BSS uptake, PN was slightly more expensive to deliver, though the cost implications were modest and we cannot give a fair evaluation of the money saved through the prevention of CRC had the PN process not been restricted.

The main challenge of this study was the recruitment of prospective BSS invitees which led to a biased, self-selected sample of highly motivated participants. SSPs suggested that the main problem lay with sending study invitations alongside the BSS preinvitations. While this was the only option in the context of an SSP-led study, it appeared to cause confusion among some patients, with an example given by one SSP of a woman assuming she was taking part in the study when she was attending her BSS appointment. Indeed, this aspect of the study design, while included to allow ease of access to appropriate patients, to avoid an additional mail out and to keep it separate from receipt of the BSS appointment letter, may still have led to information overload, negatively affecting participation rates.

This design element also related to the main obstacle for PN in the screening context, namely the lack of availability of patient telephone numbers by the screening programme. As a result, we had to ask participants to provide their telephone number during the study consent process. This meant there was a selection bias, because we were only able to contact individuals who were engaged with BSS and willing to take part in research, rather than those who were unengaged and arguably most in need of navigation (eg, those who did not read the invitation letters). SSPs commented that they felt that those who did not want to take part in BSS may have also not wanted to discuss BSS and their reasons for non-attendance with them over the phone. It was, therefore, not surprising that the uptake was much higher in our sample (79% and 80% in the control and intervention groups, respectively) compared with uptake observed within South Tyneside generally (37%) and specifically within STDH (35%) during the study period.[2] (Ritchie, personal communication, 2016).

SSPs also faced great difficulty establishing contact with participants who had consented to the study, with many unanswered calls and false telephone numbers provided. It was not possible to verify telephone numbers ahead of the PN intervention and we were not able to assess and compare the proportion of incorrect telephone numbers in the control group. While one SSP considered this a 'mischievous side' of the population, we perhaps could have benefited from asking consenting participants to provide an indication of the best day and time to contact them by phone. Additionally, as per usual practice within the centre, navigators did not leave voicemail messages, but this may have helped alert patients to a subsequent call and perhaps offered reassurance of the friendly voice they were likely to receive following their non-response or non-attendance. A request for the patient to confirm their agreement to the recording of the call before proceeding with PN may also have put people off continuing the conversation.

Future research would benefit from finding ways to contact patients without first having them provide their number and consent to being called. For example, by providing the navigation service through primary care (using healthcare assistants, nurses or volunteers trained in BSS navigation), the telephone numbers of patients will already be available and a call from their general practice may be considered a more familiar and acceptable approach. However, in a study involving telephone communication through a general practice in a socio-economically deprived area in London to increase FOBT uptake specifically, similar difficulties ensued; 46% of patients to be contacted had an incorrect number or no number at all documented.[9] Exploration of using other communication avenues to personally promote healthcare initiatives and opportunities to unengaged audiences is required.

The observation from SSP interviews that BSS awareness is low in South Tyneside suggests that an intervention to raise awareness might initially be more useful than PN in increasing BSS uptake in future. It is possible that PN is better suited to patients who are already aware of an available opportunity or concept. Alternatively, a decision aid could perhaps be used in future, alongside or ahead of PN, to help increase patients' knowledge of BSS, including its risks and benefits, and encourage consideration of the test. A recent American study found that providing patients with a decision aid, as well as PN, led to greatly increased bowel cancer screening uptake within 6 months when compared with usual care (68% vs 27%).[17]

To conclude, PN provided by SSPs was not found to be a feasible intervention to increase BSS among patients in South Tyneside Screening Centre. This was likely due to the lack of access to patient telephone numbers, causing

a selection bias whereby mostly patients engaged with BSS participated in the research, and thus, those most in need of navigation were uncontactable. In addition to this, there were difficulties contacting patients who had consented to the study, including unanswered calls and false telephone numbers. Subsequently, the feasibility of PN as an intervention in itself could not be assessed with this population. While the delivery of PN was not possible in the present study, alternative strategies to allow an evaluation of the impact of a personalised navigation approach to help patients in England engage with bowel screening opportunities are sought.

**Author affiliations**
[1]Research Department of Behavioural Science and Health, University College London, London, UK
[2]School of Health Sciences, Faculty of Health & Medical Sciences, University of Surrey, Guildford, UK
[3]South of Tyne Bowel Cancer Screening Centre, Gateshead Health NHS Foundation Trust, Queen Elizabeth Hospital, Gateshead, UK
[4]Patient Representative, London, UK
[5]Trinity Medical Centre, South Shields, UK
[6]Department of Applied Health Research, University College London, London, UK
[7]South Tyneside NHS Foundation Trust, South Tyneside District Hospital, Tyne and Wear, UK

**Acknowledgements** Thanks are given to staff within South Tyneside NHS Foundation Trust, particularly Claire Livingston, Research Lead, for her support in the set-up of the study and Research Nurses Carly Brown and Madeleine Duffy for their management of the study database and mail-outs. Thanks are also given to the North East bowel cancer screening hub for their support in coordinating the study invitation process, and to Nick Counsell, Medical Statistician within the Cancer Research UK and UCL Cancer Trials Centre, for his statistical guidance and advice.

**Contributors** CvW conceived the study with contributions to the design from LMM, LB, HM, SM and CR. LMM oversaw the management of the study, led the development of study materials and supervised HS. HS provided support to the research nurses and specialist screening practitioners (SSPs) coordinating the study roll out, led the analysis of collected data and drafted the final report. MR and HM supervised the SSPs delivering the intervention and LB was an integral part to the patient navigation training provided to SSPs and their end of study interviews. MF prepared the initial manuscript, then collated and integrated comments and reviews for the final submission. NP and SM led the cost analysis and CR oversaw study progress within South of Tyne Screening Centre. All authors have contributed to various drafts of the manuscript and have reviewed and agreed the final submission.

**Funding** This work was supported by the National Institute for Health Research (NIHR) under its Research for Patient Benefit (RfPB) Programme (Grant Reference Number PB-PG-0613-31021).

**Disclaimer** The views expressed are those of the author(s) and not necessarily those of the NHS, the NIHR or the Department of Health.

**Competing interests** None declared.

**Patient consent for publication** Not required.

**Ethics approval** Ethical approval was obtained from NRES Committee London-Bloomsbury (letter dated 31 December 2014; REC reference: 14/LO/2308).

**Provenance and peer review** Not commissioned; externally peer reviewed.

**Data sharing statement** Data can be requested from the corresponding author.

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
