## [Reviewer comments · BMJ Open]

This paper was submitted to a another journal from BMJ but declined for publication following peer review. The authors addressed the reviewers' comments and submitted the revised paper to BMJ Open. The paper was subsequently accepted for publication at BMJ Open.

(This paper received three reviews from its previous journal but only two reviewers agreed to published their review.)

ARTICLE DETAILS

TITLE (PROVISIONAL)	Using Specialist Screening Practitioners (SSPs) to increase uptake of Bowel Scope (Flexible Sigmoidoscopy) Screening: results of a feasibility single stage phase II randomised trial
AUTHORS	Mcgregor, Lesley; Skrobanski, Hanna; Ritchie, Mary; Berkman, Lindy; Miller, Hayley; Freeman, Madeleine; Patel, Nishma; Morris, Stephen; Rees, Colin; von Wagner, Christian

VERSION 1 – REVIEW

REVIEWER	Carlo Senore SSD Epidemiologia e screening – CPO AOU Città della Salute e della Scienza, Turin, Italy
REVIEW RETURNED	20-May-2018

GENERAL COMMENTS	The authors are presenting the results of well designed experimental study, aimed to assess the impact of patient navigation in the context of a population based screening program. Given the low uptake rate of screening the question addressed in this paper is important. The detailed description of the methods and of the rationale for the protocol choices represents a strength of the paper. Also, the decision to report a negative results, should be commended: the reports is providing valuable and detailed information about the reasons which might explain why the proposed approach was not successful, which may be useful to implement more successful approaches. My main concern is however related to the fact that, as the authors recognize in the discussion, the recruitment procedure implemented in the study “led to a self-selected sample of highly motivated people” As already observed in the UK Flexiscope trial, people expressing an interest in screening are more likely to attend a screening invitation. In this context, sending the study material and the request for subject's consent to be contacted, together with the pre-invitation letter, might have favoured the response from those who were considering participating in screening. Although the actual figures for non-participants are much lower than expected, this result (i.e. self-selection) could be anticipated and some measures should have been implemented in the design phase. PN, in the study performed in other jurisdictions was not used the approach adopted in this study. As the authors mention
---

	in the discussion, implementing PN through general practices might partially overcome the barrier related to the need of obtaining subjects' consent to access their phone number. The authors might want to explain why they did not consider to have the trained SSPs in the general practices. They briefly mention some reason in the discussion, but this seems a crucial issue to be addressed. Minor points p. 3 line 39. I do not see how PN could overcome financial barriers (which is different from saying that PN can have a positive impact among subjects in the low SES groups) p.6 line 23 There are published from other jurisdictions data about the role of anticipated pain and embarrassment as barriers to FS screening
--	--

VERSION 1 – AUTHOR RESPONSE

Reviewer 1

The authors are presenting the results of well designed experimental study, aimed to assess the impact of patient navigation in the context of a population based screening program. Given the low uptake rate of screening the question addressed in this paper is important. The detailed description of the methods and of the rationale for the protocol choices represents a strength of the paper. Also, the decision to report a negative results, should be commended: the reports is providing valuable and detailed information about the reasons which might explain why the proposed approach was not successful, which may be useful to implement more successful approaches.

My main concern is however related to the fact that, as the authors recognize in the discussion, the recruitment procedure implemented in the study “led to a self-selected sample of highly motivated people”. As already observed in the UK Flexiscope trial, people expressing an interest in screening are more likely to attend a screening invitation. In this context, sending the study material and the request for subject’s consent to be contacted, together with the pre-invitation letter, might have favoured the response from those who were considering participating in screening.

Although the actual figures for non-participants are much lower than expected, this result (i.e. self-selection) could be anticipated and some measures should have been implemented in the design phase. PN, in the study performed in other jurisdictions was not used the approach adopted in this study.

Thank you for your positive feedback. The way the Bowel Scope Screening process is set up by Public Health England means that only a person’s NHS number, date of birth, name and address are needed from GP practice records in order to generate the BSS pre-invitation letter etc. Telephone numbers are only obtained by the Bowel Cancer Screening Programme (BCSP) if provided by an individual when confirming their appointment. While this meant we could have access to those who would go on to confirm and then not attend their appointment, the only way we were able to obtain telephone numbers from those who would not respond to the appointment letter, was to first ask people to consent to take part in the study. Such non-responders were an important sub-population that we did not want to miss.

We were advised by the hospital not to approach people about taking part in the study once the appointment letter had been sent so as to avoid information overload, potential for confusion and subsequent non-attendance. We therefore agreed to introduce the study 2 weeks before the appointment letter would be received (study invitation sent with the pre-invitation) and would only contact relevant people who did consent after a BSS cancellation letter was generated.

We know from the UK trial that many people who show interest then do not proceed with the test and so for this feasibility study we hoped to have enough people interested in the study who would then not show interest in the test itself. In our protocol paper, we explain that based on the UK trial we conservatively predicted that only around 40% of people invited (estimated 960 over the recruitment period) to take part in this study would consent to do so and then only 30% of consenting individuals would fail to attend their appointment. We did not foresee a study recruitment rate of only 14%; however, this was an important finding with regards to the feasibility of implementing the intervention

within the current structure of the BCSP specifically. The lead author is currently working on a study that extends from this feasibility study and uses GP practice/Primary care records as an alternative way of accessing individual telephone numbers of those eligible for BSS, without written consent.

As the authors mention in the discussion, implementing PN through general practices might partially overcome the barrier related to the need of obtaining subjects' consent to access their phone number. The authors might want to explain why they did not consider to have the trained SSPs in the general practices. They briefly mention some reason in the discussion, but this seems a crucial issue to be addressed.

This is a very good question. Specialist Screening Practitioners (SSPs) are employed within the CRC screening centres and were considered the perfect candidates for the role of navigators due to their extensive knowledge of the BCSP, their experience of talking with patients about screening, and their ability to help facilitate new appointments when required. Navigation was considered a natural, logical extension of their current role. However, as the BCSP is separate from primary care, it is not possible to have SSPs within general practices as such. To move navigation to primary care would involve more intensive navigator training for selected individuals employed within GP practices (e.g. health care assistant, nurse or volunteer), and would involve indirect links to the BSS appointment process. We have now provided an additional paragraph on page 3 which elaborates on why we chose to conduct navigation through screening centres and not primary care, and on page 12 (Discussion) we have stipulated that if offered through primary care, navigation would not be conducted by SSPs. Given the results we now believe that General Practices, with their direct access to the contact details of screening invitees, might be a suitable alternative.

Minor points

p. 3 line 39. I do not see how PN could overcome financial barriers (which is different from saying that PN can have a positive impact among subjects in the low SES groups)

The reviewer is correct. PN, in and of itself, does not entail any direct financial assistance. SSPs would have been able to address specific opportunity costs, such as scheduling alternative (e.g. out of hour) appointments for people with work or childcare commitments, giving advice on the cost of travel and parking. However, as most of these costs would be encapsulated as overcoming logistical barriers we have now removed the word financial (page 2).

p.6 line 23. There are published from other jurisdictions data about the role of anticipated pain and embarrassment as barriers to FS screening

Thank you and we have updated the reference list to include several additional papers from Canada, USA and Italy. We are also now in a position to update our own work noted as 'unpublished data'; this paper was published while the current manuscript was under review (pages 6 and 15). We are also happy to add any additional references the reviewer refers to here.

VERSION 2 – REVIEW

REVIEWER	Carlo Senore AOU Città della Salute e della Scienza SSD Epidemiologia e screening - CPO Turin, taly
REVIEW RETURNED	11-Nov-2018

GENERAL COMMENTS	The authors have adequately addressed the reviewer's comments. I have no additional comments/requests Reaching non-responders represents indeed a challenge. The strategy tested by the authors, who assessed the feasibility of involving trained staff from the screening centre, would represent an efficient option, but the results of the study are indicating that, at least in their setting (where it is necessary to get subjects' consent to have access to their phone number and to be contacted), such approach is not feasible.
--

	On the other hand, it is also true, as the authors point out, that involving health care assistants or nurses in general practices, would overcome the barriers limiting the direct access from screening staff to non-responders, but it would require substantial additional investment in training for these health professionals. Therefore the paper is offering some insight on the issues to be addressed when planning to reach non-responders
--	---